# Emergence of a Reassortant 2.3.4.4b Highly Pathogenic H5N1 Avian Influenza Virus Containing H9N2 PA Gene in Burkina Faso, West Africa, in 2021

**DOI:** 10.3390/v14091901

**Published:** 2022-08-27

**Authors:** Lalidia Bruno Ouoba, Lamouni Habibata-Zerbo, Bianca Zecchin, Giacomo Barbierato, Sandaogo Hamidou-Ouandaogo, Elisa Palumbo, Edoardo Giussani, Alessio Bortolami, Mamadou Niang, Adele Traore-Kam, Calogero Terregino, Mariétou Guitti-Kindo, Angelique Angot, Dominique Guigma, Nicolas Barro, Alice Fusaro, Isabella Monne

**Affiliations:** 1Laboratoire National d’Elevage, Ouagadougou 03 BP 907, Burkina Faso; 2Division of Comparative Biomedical Sciences, Istituto Zooprofilattico Sperimentale delle Venezie, 35020 Legnaro, Italy; 3Food and Agriculture Organization of the United Nations (FAO-UN), Emergency Centre for Transboundary Animal Diseases (ECTAD), Regional Office for Africa (RAF), Accra BP 1628, Ghana; 4Food and Agriculture Organization of the United Nations (FAO-UN), Emergency Centre for Transboundary Animal Diseases (ECTAD), Ouagadougou BP 2540, Burkina Faso; 5Food and Agriculture Organization of the United Nation (FAO-UN), 00153 Rome, Italy; 6Laboratoire de Biologie Moléculaire d’Epidémiologie et de Surveillance des Bactéries et Virus Transmis par les Aliments (LaBESTA), Secteur 28, LaBESTA s/c Université joseph Ki-Zerbo, Ouagadougou 03 BP 7021, Burkina Faso

**Keywords:** highly pathogenic avian influenza, H5N1/H9N2 reassortant virus, zoonotic potential, West Africa, Burkina Faso

## Abstract

Since 2006, the poultry population in Burkina Faso has been seriously hit by different waves of Highly Pathogenic Avian Influenza (HPAI) H5N1 epizootics. In December 2021, three distinct regions of Burkina Faso, namely, Gomboussougou, Bonyollo, and Koubri, detected HPAI H5N1 viruses in poultry. Whole genome characterization and statistical phylogenetic approaches were applied to shed light on the potential origin of these viruses and estimate the time of virus emergence. Our results revealed that the HPAI H5N1 viruses reported in the three affected regions of Burkina Faso cluster together within clade 2.3.4.4b, and are closely related to HPAI H5N1 viruses identified in Nigeria and Niger in the period 2021–2022, except for the PA gene, which clusters with H9N2 viruses of the zoonotic G1 lineage collected in West Africa between 2017 and 2020. These reassortant viruses possess several mutations that may be associated with an increased zoonotic potential. Although it is difficult to ascertain where and when the reassortment event occurred, the emergence of a H5N1/H9N2 reassortant virus in a vulnerable region, such as West Africa, raises concerns about its possible impact on animal and human health. These findings also highlight the risk that West Africa may become a new hotspot for the emergence of new genotypes of HPAI viruses.

## 1. Introduction

Avian influenza represents one of the major concerns regarding animal health. Outbreaks of highly pathogenic avian influenza (HPAI) cause huge economic losses to the poultry industries and directly affect food security and the livelihood of rural areas in developing countries. The incidence of the disease in several West African countries has greatly increased in the last two decades as a consequence of the spread of the HPAI viruses of the H5 subtype descendent of the H5N1 virus A/goose/Guangdong/1/1996(Gs/GD), which was first detected in China in 1996 [1]. One of the most outstanding characteristics of this virus is its ability to evolve through the accumulation of genetic mutations and reassortment with multiple influenza subtypes producing new viral variants [2].

The poultry sector in Burkina Faso has a leveraging role in national economic growth and in reducing food insecurity and poverty. Since 2006, the poultry population in Burkina Faso has been seriously hit by different waves of Highly Pathogenic Avian Influenza (HPAI) H5N1 epizootics, introduced by migratory birds or by the cross-border trade of contaminated live poultry and poultry products [1]. After two serious HPAI epizootics in 2006 and 2015, the country declared that it was free from notifiable Highly Pathogenic Avian Influenza virus in 2017 [3]. In 2021, six years after the last HPAI event, Burkina Faso and other West African countries were involved in a new dramatic outbreak of the HPAI H5N1 virus [4,5,6].

The detection of HPAI H5N1 virus in a country, where viruses of the H9N2 subtype have been widely circulating since the beginning of 2017 [7], is a cause for concern not only for animal health implications and economic drawbacks, but also for the possible emergence of reassortant viruses with unknown biological properties. 

This study reports the results obtained from the characterization of the complete genome of three H5N1 viruses identified in December 2021 in Burkina Faso and describes the emergence of a novel natural reassortant highly pathogenic H5N1 influenza virus containing the H9N2 polymerase acidic (PA) gene in West Africa.

## 2. Materials and Methods

### 2.1. Case History

In December 2021, following the increase in mortality in three farms rearing different poultry species (Table 1) located in three different areas of Burkina Faso, tracheal and cloacal swabs were collected from dead and sick birds and sent to the Laboratoire National d’Elevage in Ouagadougou for a preliminary diagnosis. H5N1-positive samples were then submitted to the WOAH (previously OIE)/FAO Reference laboratory for Avian Influenza in Italy (Istituto Zooprofilattico Sperimentale delle Venezie, IZSVe) for the confirmatory diagnosis and genetic characterization of the identified viruses.

### 2.2. Virus Identification and Characterization 

Total RNA was extracted using the QIAamp Viral RNA Mini Kit (Qiagen, Hilden, Germany) according to the manufacturer’s instructions. 

The detection of Influenza A (M-gene) was performed using real-time RT-PCR (rRT-PCR) [8] on three tracheal and three cloacal samples using AgPath-ID™ One-Step RT-PCR Reagents (Applied Biosystems™, Waltham, Massachusetts, USA). Positive samples were tested for the H5 subtype through rRT-PCRs [9,10] as well as one-step RT-PCRs [11] in order to identify the pathotype of the hemagglutinin cleavage site via Sanger sequencing. Neuraminidase (NA) typing was performed through rRT-PCR, using multiple oligonucleotide sets based on the assay developed by Hoffmann et al. [12].

The amplification of the complete genomes from the tracheal samples for sequencing on the Illumina MiSeq platform was performed using the SuperScript III One-Step RT-PCR kit and Platinum Taq High Fidelity kit (Invitrogen, Carlsbard, CA, USA) according to the protocol suggested by Zhou et al. (2009) [13], which was slightly modified as previously described [1]. The Illumina Nextera DNA XT Sample preparation kit was used to prepare sequencing libraries, which were pooled in equimolar concentrations and underwent 250 bp paired-end sequencing using Illumina MiSeq in multiplex (Illumina, San Diego, CA, USA), according to the manufacturer’s instructions.

Illumina reads quality was assessed using FastQC v0.11.2 (https://www.bioinformatics.babraham.ac.uk/projects/fastqc/, accessed on 11 January 2022). Raw data were filtered by removing reads with more than 10% of undetermined (“N”) bases, reads with more than 100 bases with Q score below 10, and duplicated paired-end reads. Remaining reads were clipped from Illumina adaptors Nextera XT with scythe v0.991 (https://github.com/vsbuffalo/scythe, accessed on 11 January 2022) and trimmed with sickle v1.33 (https://github.com/najoshi/sickle, accessed on 11 January 2022). Reads shorter than 80 bases were discarded.

High-quality reads were aligned against reference sequences using BWA v0.7.12 (https://github.com/lh3/bwa, accessed on 11 January 2022) [14]. In order to correct potential errors, realign reads around indels and recalibrate the base quality, the alignment was processed using Picard-tools v2.1.0 (http://picard.sourceforge.net, accessed on 11 January 2022) and GATK v3.5 (https://github.com/moka-guys/gatk_v3.5, accessed on 11 January 2022) [15,16,17]. LoFreq v2.1.2 (https://github.com/CSB5/lofreq, accessed on 11 January 2022) [18] was run on fixed alignment to produce vcf files containing both SNPs and indels. The consensus sequences were submitted to the GISAID EpiFlu™ database (http://www.gisaid.org) under the accession numbers EPI2053416–EPI2053439 (Table 1).

The H5N1 strain (A/avian/Burkina_Faso/21VIR11911-3/2021) was isolated by inoculating 9-to-11-day-old embryonated specific pathogen-free hen eggs via the allantoic cavity, and its pathogenicity was determined by estimating the intravenous pathogenicity index (IVPI) in six-week-old specific pathogen-free (SPF) chickens, according to standard procedures as per the World Organization for Animal Health [19].

### 2.3. Phylogenetic and Evolutionary Analyses

Representative sequences of recent H5N1 viruses and sequences resulting from the BLAST search of the sequences obtained in this study were retrieved from the GISAID database (http://www.gisaid.org, accessed 1 June 2022). Sequences were aligned in MAFFT v7 [20]. Maximum likelihood phylogenetic trees were generated in IQTREE v1.6.6 (https://github.com/iqtree/iqtree1, accessed on 11 January 2022) [21,22], performing an ultrafast bootstrap resampling analysis (1000 replications). Phylogenetic trees of each gene segment were visualized in FigTree v1.4.4 (http://tree.bio.ed.ac.uk/software/figtree/, accessed on 11 January 2022).

A Markov chain Monte Carlo (MCMC) analysis of the HA gene was performed using BEAST v1.10.4 in combination with the BEAGLE libraries [23]. A strict molecular clock and the SRD06 substitution model were used along with a Constant Size coalescent tree prior as previously described [24]. MCMC chains were run for 50 million iterations and convergence was assessed using Tracer v1.7.1 [24,25,26,27]. The MCC tree was summarized using TreeAnnotator v1.10.4 (http://beast.bio.ed.ac.uk/TreeAnnotator/, accessed on 25 March 2022) and visualized in FigTree v1.4.4 (http://tree.bio.ed.ac.uk/software/figtree/, accessed on 25 March 2022).

## 3. Results

### 3.1. Case History

In early December 2021, a dramatic increase (up to 90%) in mortality was reported by poultry producers from three poultry farms located in three distinct municipalities and regions of the country (Gomboussougou, Bonyollo, and Koubri) (Table 1, Figure 1).

The three municipalities are geographically isolated, and located within an average radius of 100 km from the capital Ouagadougou, but they share a significant development of poultry farming activities. The first infected farm located in Gomboussougou was a semi-open building farm rearing free-range chickens, guinea fowls, turkeys and geese; the second one in Bonyollo was a traditional farm, with free-range poultry and precarious shelter; the last one in Koubri was a closed-house chicken farm with an automatic air extraction system. The sudden death of the birds, which was sometimes associated with the appearance of nervous signs, such as torticollis and incoordination, was observed in all the affected farms. In Gomboussougou and Bonyollo, mortality events occurred almost in the same period (7 December 2021 and 8 December 2021), while in Koubri, the deaths started a week later, on 16 December 2021. Postmortem examination revealed the congestion of the internal organs and a bloody discharge from the orifices. 

### 3.2. Virus Identification and Characterization

All the tested samples from Burkina Faso (three tracheal and three cloacal samples) were positive for the Highly Pathogenic Avian Influenza virus.

Whole genome sequences were generated from three H5N1 samples collected from the three affected farms. The analysis of the complete hemagglutinin (HA) gene segment showed that the HPAI H5N1 viruses from Burkina Faso belonged to clade 2.3.4.4b. The three viruses clustered together (similarity of 99–99.2% for the HA gene) and with HPAI H5N1 viruses identified in Nigeria in 2021 (similarity of 98.5–99.1% for the HA gene) and in Niger in 2022 (similarity of 98.1–98.4% for the HA gene) (Figure 2). The analyses of the other gene segments showed that the H5N1 viruses from Burkina Faso were closely related to H5N1 viruses from Nigeria, Niger and Senegal (2020–2022) (Appendix A, Appendix A) except for the PA gene (Figure 2), which clusters with H9N2 viruses of the G1 lineage collected in West Africa between 2017 and 2020. 

The estimation of the time to the most recent common ancestor (tMRCA) performed on the HA gene indicated that this virus might have emerged in early June 2021 (95% HPD, March-August 2021) (Appendix A, Appendix A).

These new reassortant viruses from Burkina Faso present a highly pathogenic cleavage site (KRRKRGLF) similar to that of H5N1 viruses circulating in West Africa and Europe, and possess several mutations that are likely associated with an increased zoonotic potential or adaptation to poultry (Table 2).

The three viruses possess the HA-S137A (H3 numbering) mutation in the HA protein, which causes an increased alpha 2,6-SA binding. In general, avian-adapted Influenza A viruses preferentially bind SA α2-3 Gal, and human-adapted IAVs have a binding preference for SA α2-6 Gal [28]. This mutation had previously been observed in the African and European H5N1 viruses collected in the period 2020–2021. The viruses under study present a twenty-two-amino acid deletion in the NA stalk region, similarly to the H5N1 viruses identified in Nigeria and Niger in the period 2021–2022. It has been shown that this feature decreases the ability of NA to release the virus from the cells [36,37] and increases the virulence of the virus for chickens [38]; in addition, the deletion in the stalk of the NA gene is a marker of virus adaptation from wild aquatic birds to poultry [33]. Furthermore, the H5N1 virus collected from the third outbreak in Koubri presents a deletion in the NS1 protein at positions 78–80. This three-amino acid deletion is positioned immediately before the five-amino acid deletion described by Li et al., 2014 [39]. The author demonstrated that viruses with both a deletion of 20 amino acids in the stalk of the neuraminidase (NA) glycoprotein and a deletion of five amino acids at positions 80 to 84 in the non-structural protein NS1 contribute to the high pathogenicity of H5N1 avian influenza viruses (AIVs) in ducks [39]. However, the effect of a three-amino acid deletion in the NS1 protein requires further investigation.

The IVPI value of the virus A/avian/Burkina_Faso/21VIR11911-3/2021 was 3.0, thus confirming that the isolate can be considered highly pathogenic according to the definition of the World Organisation for Animal Health [19].

## 4. Discussion

The result of the epidemiological investigations, which indicates separate viral introductions into the three affected farms, and the genetic similarity characterizing these reassortant viruses, may suggest that a single reassortment event had occurred in an unspecified location in West Africa. The virus later evolved independently and most probably emerged in Burkina Faso in early June 2021, as per the Bayesian analysis. Despite the absence of a proven epidemiological link between the three outbreaks, informal exchanges of animals between the different regions may have occurred. The period in which the influenza outbreaks were reported corresponds to a time of the year characterized by many social events (weddings, engagements, etc.) involving the movement of poultry (purchases and donations), thus allowing the rapid spread of the virus. Additionally, the three regions in which the farms are located host wild bird gathering areas as well as forest reserves, which constitute significant risk factors. However, the close genetic relationship with the H5N1 viruses circulating in poultry in West Africa since early 2021, as well as the reassortment event with the H9N2 subtype of the G1 lineage rarely observed in wild animals, suggests that poultry represent the most likely source of emergence of this strain.

Despite the remarkable ability of clade 2.3.4.4 H5Nx HPAI viruses to reassort [45] and the extensive circulation of both clade 2.3.4.4 H5Nx and H9N2 viruses in Asian and African poultry, reassortment events between H9N2 and HP H5Nx viruses of clade 2.3.4.4 have only rarely been reported. In particular, natural HP 2.3.4.4 H5Nx/H9N2 reassortant viruses were detected in domestic birds in China (2015, 2016) [46,47,48] and Egypt (2018, 2019) [49,50]. It is relevant to highlight that 2.3.4.4 HPAI H5N6 viruses with internal gene constellations derived from H9N2 viruses had also infected humans in 2015 and 2016 in China [51].

## 5. Conclusions

Although it is difficult to ascertain where and when the reassortment event occurred, the emergence of this H5N1/H9N2 reassortant virus in West Africa raises concerns about its possible impact on animal and human health. Further studies are urgently required to evaluate the biological significance of this emerging genotype to investigate if this reassortment may increase the zoonotic potential of H5N1 viruses and to assess its spread.

The massive global circulation of HPAI H5N1 viruses in wild and domestic birds threatens a future scenario characterized by the possible continuous emergence and re-emergence of new viral variants in the African continent. Long-term control measures intended to improve high-risk industry practices, such as biosecurity measures and live poultry market hygiene, and means to strengthen the early detection and genetic characterization of AIV, must be implemented in West Africa to reduce animal and public health risks.

## Figures and Tables

**Figure 1 viruses-14-01901-f001:**
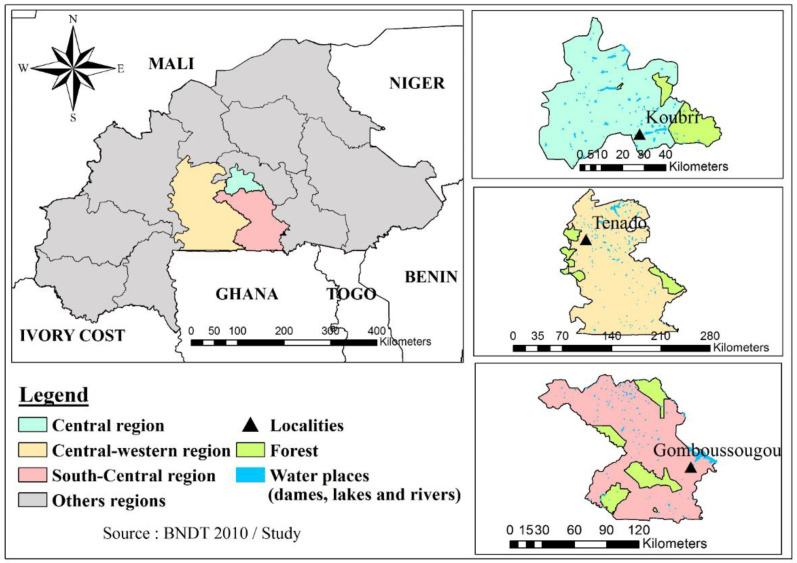
Map of the farms in Burkina Faso. The three farms are located in three different areas: central region (in light blue), central–western region (in light orange), and southern–central region (in light pink). For each of the three regions, localities are indicated by triangles, forests are indicated in green, and water resources are shown in blue.

**Figure 2 viruses-14-01901-f002:**
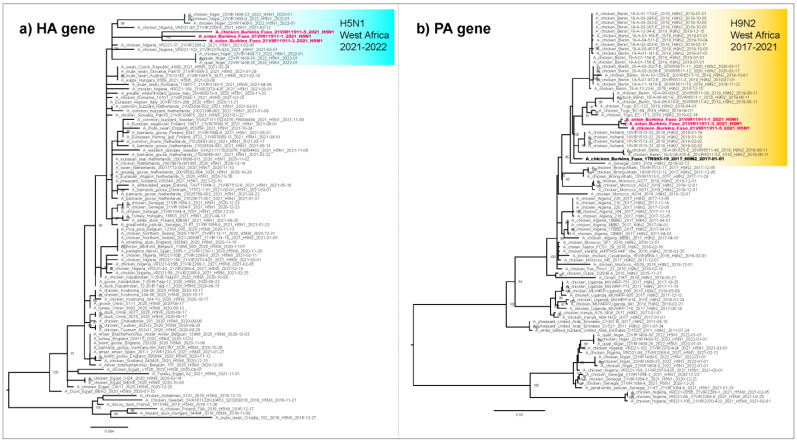
Maximum likelihood phylogenetic trees of the hemagglutinin (HA) and polymerase acidic (PA) genes. (**a**) Maximum likelihood phylogenetic tree of the HA gene. The light blue box shows the cluster of H5N1 viruses identified in West Africa in the period 2021–2022; the HPAI H5N1 viruses from Burkina Faso are highlighted in bold pink. (**b**) Maximum likelihood phylogenetic tree of the PA gene. The yellow box shows the cluster of H9N2 viruses identified in West Africa in the period 2017–2021; the HPAI H5N1 viruses from Burkina Faso are highlighted in bold pink; the 2017 H9N2 virus from Burkina Faso is highlighted in bold black. The trees were inferred using IQTREE v1.6.6. Ultrafast bootstrap supports higher than 75 are indicated next to the nodes.

**Table 1 viruses-14-01901-t001:** Epidemiological information.

Sites	Number of Animals	Species	Sample ID	Mortality Rate	Sampling Date	Sequenced Virus	Accession Number (GISAID)
Central–Western Region (Bonyollo)	320	Chickens, Guinea fowls, Pigeons	BKF TR 1	100%	08 Dec 2021	A/avian/Burkina_Faso/21VIR11911-1/2021	EPI2053416-EPI2053423
Southern–Central Region (Gomboussougou)	1143	Chickens, Guinea fowls, Turkeys, Geese	BKF TR 2	97,63%	07 Dec 2021	A/avian/Burkina_Faso/21VIR11911-3/2021	EPI2053424-EPI2053431
Central Region (Koubri)	210,000	Chickens	BKF TR 3	6,13%	16 Dec 2021	A/chicken/Burkina_Faso/21VIR11911-5/2021	EPI2053432-EPI2053439

BKF: Burkina Faso.

**Table 2 viruses-14-01901-t002:** Mutations identified in the HPAI H5N1 viruses from Burkina Faso.

Protein	Mutation	HPAI H5N1 Viruses Burkina Faso	Effect	Reference
HA	S137A, H3 numbering (S149A from the initial Met)	All 3 viruses	Increased pseudovirus binding to α2–6.	[28]
HA	HA-A158T, H3 numbering (A172T from the initial Met)	All 3 viruses	Creates a new potential N-glycosylation site. The motif at positions 156–158 changed from NDA to NDT.	n.a.
PB2	T105V and A661T	All 3 viruses	Host specificity markers identified through statistical methods (T in avian, V in human for T105V and A in avian, T in human for A661T). The mutations lie in regions of PB2 both known for binding to PB1 and NP.	[29]
PA	L268I and S409N	All 3 viruses	Host specificity markers identified through statistical methods (L in avian, I in human for L268I and S in avian, N in human for S409N).	[30]
NA	NA stalk deletion	All 3 viruses, similarly to the H5N1 viruses identified in Nigeria in 2021	A deletion in the stalk region of the NA decreases the ability of NA to release the virus from cells and increases the virulence of the virus in mice and chickens. In addition, it is a marker of virus adaptation from wild aquatic birds to poultry.	[31,32,33,34,35,36,37,38]
NS1	Deletion at positions 78–80	A/chicken/Burkina_Faso/21VIR11911-5/2021	This 3-amino acid deletion overlaps with a deletion of five amino acids previously described [39]. Viruses with both a deletion of 20 amino acids in the stalk of the NA glycoprotein and a deletion of 5 amino acids at positions 80 to 84 in the NS1 protein contribute to the high pathogenicity of H5N1 AIVs in ducks.	[39]
NS1	P42S	All 3 viruses	Increased virulence in mice.	[40]
NS1	P87S	All 3 viruses	P87S is a host specificity marker identified through statistical means (S in human, P in avian).	[41]
NS1	103F	All 3 viruses	Increased virulence in mice.	[42,43]
NS2	T47A (with NS1-205S)	A/chicken/Burkina_Faso/21VIR11911-5/2021	Decreased antiviral response in ferrets.	[44]

HA: hemagglutinin; PB2: polymerase basic 2; PA: polymerase acidic; NA: neuraminidase; NS1: non-structural 1; NS2: non-structural 2; n.a.: not available; AIVs: avian influenza viruses.

## Data Availability

The consensus sequences of the viruses analyzed in this study were submitted to the GISAID EpiFlu™ database (http://www.gisaid.org accessed on 25 August 2022) under the accession numbers EPI2053416–EPI2053439 (details in Table 1).

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
