# Peer review of "Emergence of a Reassortant 2.3.4.4b Highly Pathogenic H5N1 Avian Influenza Virus Containing H9N2 PA Gene in Burkina Faso, West Africa, in 2021"

_viruses, 2022, doi:10.3390/v14091901_

Round 1

Reviewer 1 Report

Lalidia-Ouoba and colleagues present a study on a recent discovered reassortant that has taken up a new segment with potential zoonotic properties based on a virus already circulating in Africa. Three outbreaks in Burkina Faso are described in detailed case reports, geographical overviews and phylogenetic analyses. The study is brief but precise and describes both the epidemiological case investigations and a detailed genetic characterisation. The chosen publication form as communication therefore fits very well. The following minor corrections should be considered:

Title: As the three described cases emerge in Burkina Faso this should be mentioned in the title as well as the time of emergence. This stresses the actuality of the study e.g. "Emergence of a reassorted 2.3.4.4b highly pathogenic H5N1 avian influenza virus containing H9N2 PA gene in Burkina Faso, West Africa in 2021"

lines 158ff.: It should be added for which segment the % identities apply. Do similartities vary between segments e.g. for the M or NS segments?

Figure 2: Please correct the name for sample 21VIR11911-5 (from avian to chicken) to be consistant with text/tables and trees in the supplement and the naming in the repository submission.

Table 2: Same correction as above necessary for 21VIR11911-5

Author Response

Reviewer 1

Lalidia-Ouoba and colleagues present a study on a recent discovered reassortant that has taken up a new segment with potential zoonotic properties based on a virus already circulating in Africa. Three outbreaks in Burkina Faso are described in detailed case reports, geographical overviews and phylogenetic analyses. The study is brief but precise and describes both the epidemiological case investigations and a detailed genetic characterisation. The chosen publication form as communication therefore fits very well. The following minor corrections should be considered:

Author’s response. We thank the reviewer for his kind comments. We have answered the question and modified the title, figure 2 and table 2 according to the reviewer’s suggestions. We have also provided an introductory paragraph to contextualize the HPAI H5 virus, a brief description of previous HPAI H5 epidemics in Burkina Faso and relevant references in the introduction section.

Title: As the three described cases emerge in Burkina Faso this should be mentioned in the title as well as the time of emergence. This stresses the actuality of the study e.g. "Emergence of a reassorted 2.3.4.4b highly pathogenic H5N1 avian influenza virus containing H9N2 PA gene in Burkina Faso, West Africa in 2021".

Author’s response. We thank the reviewer for his suggestion. We modified the title as follows: “Emergence of a reassortant 2.3.4.4b highly pathogenic H5N1 avian influenza virus containing H9N2 PA gene in Burkina Faso, West Africa, in 2021”.

Lines 158ff.: It should be added for which segment the % identities apply. Do similarities vary between segments e.g. for the M or NS segments?

Author’s response. We specified that the similarity is reported for the HA gene segment (lines 183-185). The similarities among the viruses from Burkina Faso and viruses from Nigeria, Niger and Senegal are comparable for all the gene segments (98.9%-99.4% similarity for the PB2 gene; 98.9%-99.2% for the PB1 gene; 98.9%-99.1% for the NP gene; 98.1%-99.1% for the NA gene; 98.4%-99.1% for the M gene; 99.1%-99.5% for the NS gene).

Figure 2: Please correct the name for sample 21VIR11911-5 (from avian to chicken) to be consistent with text/tables and trees in the supplement and the naming in the repository submission.

Author’s response. We apologize for the oversight. We have amended accordingly.

Table 2: Same correction as above necessary for 21VIR11911-5.

Author’s response. We apologize for the oversight. The correction has been made.

Author Response

Reviewer 2

Bruno Lalidia-Ouoba et al in their manuscript report the Emergence of a reassortant 2.3.4.4b highly pathogenic H5N1 2 avian influenza virus containing H9N2 PA gene in West Africa and highlight the risk that West Africa may become a new hotspot for the emergence of new genotypes of HPAI viruses. The HPAI H5N1 viruses found in the three affected regions of Burkina Faso were phylogenetically clustered together within clade 2.3.4.4b, and are closely related to HPAI H5N1 viruses identified in Nigeria and Niger in 2021-2022. They also identified the PA gene in these HPAI H5N1 viruses clustered with H9N2 viruses of the zoonotic G1 lineage collected in West Africa between 2017 38 and 2020. Additionally, several mutations likely associated with an increased zoonotic potential were found in these reassortant viruses.

Many minor mistakes were found in the manuscript in the following.

Author’s response. We thank the reviewer for his comments and we apologize for the inaccuracies. We have now modified all the sentences according to the reviewer’s suggestions. We have also provided an introductory paragraph to contextualize the HPAI H5 virus, a brief description of previous HPAI H5 epidemics in Burkina Faso and relevant references in the introduction section.

  1. Line 65: change “…..species (Table 1). Located…” to “……species (Table 1) located….”

Author’s response. We apologize for the oversight that has now been amended.

  1. Line 76: “The detection of Influenza A (M-gene) was …”  to “Detection of Influenza A (M-gene) was ….”

Author’s response. We thank the reviewer for noticing the inaccuracy. It has now been changed accordingly.

  1. Line 76: change “….on tree tracheal and three cloacal samples…” to “: ….on three tracheal and three cloacal samples…”

Author’s response. We thank the reviewer for noticing the inaccuracy. It has been amended accordingly.

  1. Line 79: “….for the H5 subtype by RRT-PCRs [3,4] …”for the H5 subtype by RT-PCRs [3,4]….”

Author’s response. We apologize for the oversight, we have replaced “RRT-PCRs” with “rRT-PCRs” (real-time RT-PCRs, as specified in line 76).

  1. Line 83 “…for the sequencing…” to “…for sequencing….”

Author’s response. We thank the reviewer for noticing the inaccuracy. We have changed it accordingly.

  1. Line 123-124: “We employed a strict molecular clock and the SRD06 substitution model was used along with a Constant Size coalescent tree prior [19].” To “A strict molecular clock and the SRD06 substitution model were used along with a Constant Size coalescent tree as described previously [19].

Author’s response. We have changed the sentence following the reviewer’s suggestions.

  1. Line 153-154: “….resulted positive for…” to “…were positive for….”

Author’s response. We have modified accordingly.

  1. Line 157: “shows that the HPAI H5N1 viruses from Burkina Faso belong to clade 2.3.4.4b” to “showed that the HPAI H5N1 viruses from Burkina Faso belonged to clade 2.3.4.4b”

Author’s response. We thank the reviewer for noticing the inaccuracy that has now been amended according to the reviewer’s suggestion.

  1. Line 158: “viruses cluster together (similarity of 99%-99.2%) and also with HPAI H5N1…” to “viruses clustered together (similarity of 99%-99.2%) with HPAI H5N1….”

Author’s response. We thank the reviewer for noticing the inaccuracy, which has now been amended.

  1. Line 160-161:” The analyses of the other gene segments show that the H5N1 160 viruses from Burkina Faso are closely related to H5N1 viruses….” To “Analyses of the other gene segments showed that the H5N1 160 viruses from Burkina Faso were closely related to H5N1 viruses….”

Author’s response. We thank the reviewer for noticing the inaccuracy. We have now modified accordingly.

  1. Line 174:” The estimation of the time…” to “Estimation of the time….”

Author’s response. We thank the reviewer for noticing the inaccuracy. We have changed it accordingly.

  1. Line 179:”… mutations likely associated to an increased…” to “…. mutations likely associated with an increased….”

Author’s response. We thank the reviewer for noticing the inaccuracy. It has been modified accordingly.